# Experimental Study on Frost Crystal Morphologies and Frosting Characteristics under Different Working Pressures

**Tianwei Lai \*, Xue Liu, Xiaojun Dong, Mingchen Qiang, Shaohang Yan and Yu Hou**

State Key Laboratory of Multiphase Flow in Power Engineering, Xi'an Jiaotong University, Xi'an 710049, China;
liuxuewith@stu.xjtu.edu.cn (X.L.); miller@allfavorpcb.com (X.D.); qmc990223@stu.xjtu.edu.cn (M.Q.);
ysh1040014659@stu.xjtu.edu.cn (S.Y.); yuhou@mail.xjtu.edu.cn (Y.H.)
**\*** Correspondence: laitianwei@mail.xjtu.edu.cn; Tel.: +86-15-2029-52214

**Abstract:** Frost usually grows on the cooling surface of heat transfer equipment, which often operates under different working pressures. In such conditions, one of the fundamental influencing factors of frosting, the diffusion coefficient of water vapor around the cooling surface, is different from that in the normal atmosphere pressure. In order to investigate the pressure effect on frosting, a test rig for frosting visualization is designed. The role of working pressure around the cooling surface on the frost crystal morphologies and frost layer characteristics is evaluated under different air parameters and temperatures of cooling surface. The vertical growth and lateral distribution of frost crystals are strongly related to working pressure around the cooling surface. More frost crystal morphologies and faster growth rate of frost layer are presented at a lower pressure. The frost thickness, frost accumulation and density reduce with higher pressure. The dimensionless correlation of frost thickness is proposed considering working pressure around the cooling surface, humid air parameters, and temperature of cooling surface.

**Keywords:** frosting; pressure; frost crystal morphology; frost layer characteristic; correlation





## 1. Introduction

In nature and industry, frost usually grows at normal atmospheric pressure ($1.013 \times 10^5$ Pa) [1–4]. Whereas in some industrial fields and high-altitude areas, frosting also occurs at the ambient above or below atmospheric pressure. For example, a high-pressure and low-temperature environment are required in the cold working process for sterilization, so that frost inevitably develops on the processed product [5]. In addition, frost also forms at low-pressure and low-temperature areas, such as astronomical detection instruments and air pre-coolers in aircraft [6]. In these devices, the operational efficiency and even safety are deteriorated due to frost formation.

The frost growth consists primarily of several periods: (1) crystal growth period, (2) frost layer growth period, and (3) frost layer full growth period [7]. In different periods, the frost thickness and frost density, which represent the most important features of frost, are influenced prominently by the frost crystal morphologies and frost layer characteristics [8]. As for the frost crystal, the nucleation and growth of frost crystals are essentially the result of the water vapor diffusion from the humid air to the cooling surface. From the perspective of phase change, there are two types of frost crystal formation: (1) water vapor—water droplet—frost crystal, and (2) water vapor—frost crystal, directly [9]. In the first type of frosting, the water vapor in humid air precipitates and condensation appears when the temperature of the cooling surface is lower than the dew point temperature and 0 °C. There is a certain amount of supercooling degree on the frozen droplets. A sharp bulge is formed on the top of the completely frozen droplet due to the expansion from water to ice. After that, the frost crystals develop some vertical column crystals, which are spread out without affecting each other [10,11]. In this process, the frost crystals may collapse and melt. In the second type of frosting, the water vapor in the humid air directly condenses into frost

crystals, when the temperature and the partial pressure of water vapor are lower than the triple point of water (273.16 K, 610.75 Pa). In the crystal growth period, different frost crystal morphologies are developed under different ambient conditions, such as column, plate and dendritic [12]. Moreover, irregular, scaly, and plate shapes may also appear [13]. The frost crystal morphologies are mainly influenced by the temperature of air and cooling surface [7,14,15]. In addition, when the temperature of frost is higher than −18 °C and the water vapor supersaturation around the frost crystals is large enough, the growth habit of the frost crystals is principally affected by the water vapor supersaturation in the environment [16]. In addition, in an ambient environment with low or high pressure, the diffusion coefficient of water vapor is different from that in the normal atmosphere pressure [17]. Therefore, there exists some differences in the growth habits of frost crystals.

As for the frost layer, the growth characteristics depend primarily on the crystallization speed. In the frost layer growth period, the frost crystals start to grow in both vertical and lateral directions [18]. During this period, the gaps between the frost crystals are filled with newly grown frost crystals. Therefore, the variation of frost characteristics is significant. Moreover, frosting is a phase change process, which is restricted by a variety of factors. Most research on frosting characteristics is conducted by adjusting the temperature, humidity ratio, velocity of the humid air, and temperature of the cooling surface [19–21]. Among them, the humidity ratio and the temperature of the cooling surface lead to a greater influence. The effect of humidity ratio on the frosting is essentially driven by the partial pressure difference of water vapor. The growth of the frost layer is faster with a higher partial pressure of water vapor [22]. Under the same air temperature, humidity ratio, and cold surface temperature, a thinner and denser frost layer under lower atmospheric pressure is found [23].

In terms of frost crystal morphology and frost layer growth, the supersaturation of the humidity ratio on the frost surface has been studied by many researchers [8]. The humidity ratio is a function of overall ambient pressure, partial pressure of water vapor, and temperature of humid air. These parameters can be utilized to investigate the pressure effect on frosting characteristics independently. Among the three parameters, the influences of partial pressure of water vapor and temperature of humid air are well comprehended, while the role of overall pressure of air is usually ignored. The overall pressure can be regarded as one of the key factors affecting the diffusion of water molecules. Whereas the frost growth habits in high and low working pressure are still unclear enough today. Therefore, revealing the role of overall pressure of the air in frost growth habits is an important prerequisite for enhancing the efficiency and safety of the heat transfer equipment. In order to investigate the frost crystal morphologies and the frost characteristics under different working pressures around the cooling surface, a test rig is designed for frosting visualization. In the test rig, a visualized cavity is built with adjustable pressure. In addition, the partial pressure of water vapor, temperature of humid air, and temperature of cooling surface in the cavity are independently controlled for studying the frost characteristics. The frost crystal morphology, frost thickness, frost accumulation, and frost density are obtained under different working pressures around the cooling surface. For predicting the pressure effect on frosting, a dimensionless correlation of frost thickness is proposed, considering working pressure around the cooling surface, partial pressure of water vapor, temperature of humid air, and temperature of cooling surface.

## 2. Experimental Equipment and Data Processing

### 2.1. Frosting Visualization Test Rig with Adjustable Pressure

The details of the frosting visualization test rig are shown in Figure 1. The test rig is composed of four parts: the humid air conditioning system, the frosting visualization cavity, the frosting image acquisition system, and the cascade refrigeration system. The target temperature of air, partial pressure of water vapor, and overall pressure in the frosting visualization cavity are mainly provided by the humid air conditioning system. The humid air conditioning system is mainly made up of compressors, freeze dryers, electric

proportional valves, gate valves, buffer tanks, vacuum pumps and chillers. The goal of this research is to figure out the effect of working pressure around cooling surface on frosting. Therefore, the internal pressure of the visualized cavity needs to be adjusted steadily from low to high. With regard to high pressure, for maintaining the internal pressure of the cavity higher than atmospheric pressure, the compressor in this experiment compresses the ambient air to 0.45 MPa. Then the high pressure air is supplied into the buffer tank I to stabilize the air pressure. Two streams are directed from buffer tank I. One stream is directed into the freeze dryer and molecular sieve to remove water vapor and tiny particles in the humid air to obtain dry air. The other stream of humid air is directed into buffer tank II. By controlling the different mixing proportions of humid air and dry air, the partial pressure of water vapor can be adjusted. As for low pressure conditions, the pressure in the cavity is maintained by a vacuum pump. Buffer tank III is arranged between the vacuum pump and the visualized cavity to expand the pressure range. The temperature of air is adjusted by heat transfer between the chiller and the air at the outlet of buffer tank II.

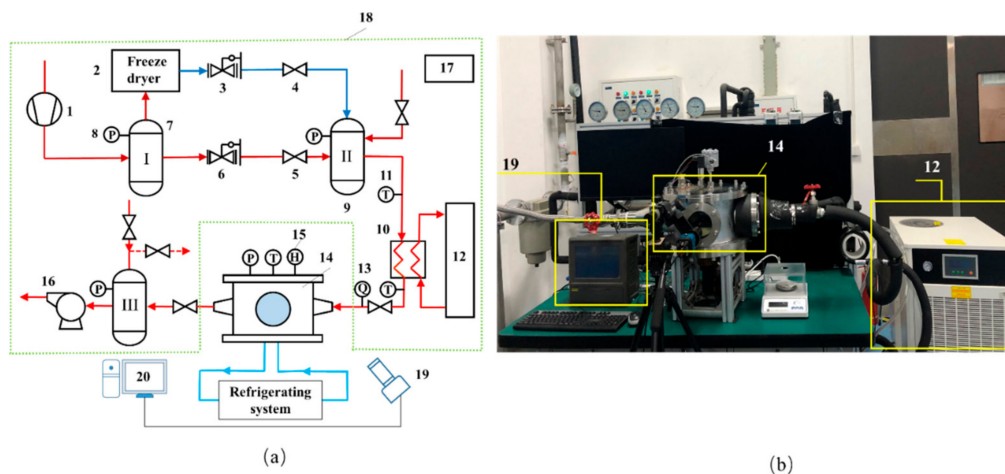

**Figure 1.** (**a**) The flowchart and (**b**) test rig of the visualized pressure adjustable frosting test, which consists of 1. compressor, 2. freeze drying equipment, 3. electropneumatic proportional valve, 4. gate valve, 5. gate valve, 6. electropneumatic proportional valve, 7. buffer tank I, 8. pressure transducer, 9. buffer tank II, 10. heat exchanger, 11. temperature sensor, 12. chiller, 13. flowmeter, 14. visualized cavity, 15. hygrometer, 16. vacuum pump, 17. atmospheric environment, 18. humid air conditioning system, and 19. data collecting system.

The frost formation is on the cooling surface inside the frosting visualization cavity, as shown in Figure 2. The test section consists of refrigerant flow, cooling surface of copper, thermocouples, and the heat insulator. The stable low temperature required for frosting on the cooling surface is provided by the injection of refrigerant flow from the cascade refrigeration system to the cooling surface. The cooling surface is a circular plane with diameter of 35mm. In order to maintain a uniform temperature distribution, a copper material with good thermal conductivity is selected as the test surface. With the aim of reducing heat loss, a thermal insulating material is wrapped around the outside of the cooling surface. On the basis of these conditions, the temperature of the cooling surface is measured by four thermocouples that are distributed around the circumferential side of the cooling surface. Moreover, the frost crystal morphologies and frost layer characteristics on the cooling surface are obtained through the image acquisition system. In addition, the CCD camera (GaoPin GP-660V) captures frosting in the direction perpendicular to normal to the cooling surface. The air temperature inside the cavity is measured by thermocouple hanging at the entrance of the cavity.

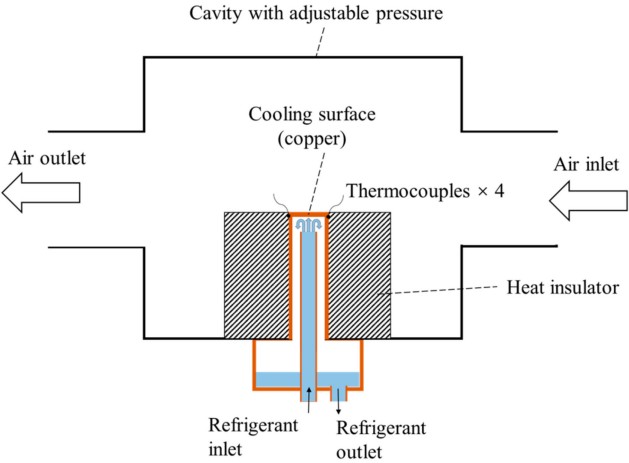

**Figure 2.** Schematic diagram of test cavity with adjustable pressure.

*2.2. Data Processing*

In the experiment, the pressure around the cooling surface, partial pressure of water vapor in cavity, temperature of air, and the temperature of cooling surface are recorded every 5 s. The frost layer image is collected every 1 min. The frost image taken by the CCD camera is binarized to obtain the average frost thickness. After the experiment, the cooling surface is heated with a heater until all the frost melts into water. The frost accumulation is obtained by weighing the melted frost, which is absorbed by the dust-free paper. In this paper, the frost density is calculated indirectly using the frost accumulation and the average frost thickness, which is shown in Equation (1):

$$\rho_f = m/(y_f \cdot A) \tag{1}$$

The uncertainties of the frost thickness and frost accumulation are calculated using the method of Lee et al. [24]. The uncertainty of the frost density is calculated indirectly by the uncertainties of frost thickness and frost accumulation. The accuracy of the apparatus used and calculated variables in this test are shown in Table 1.

**Table 1.** Uncertainty of measured and calculated variables.

| Apparatus (Variables) | Accuracy (Uncertainty) |
| --- | --- |
| Pressure transmitter | ±0.3% |
| Temperature and humidity sensor | ±0.1 °C (Temperature) ±1.5% (Relative humidity) |
| Thermocouples | ±0.5 °C |
| Frost thickness | ±4.04% |
| Frost accumulation | ±4.16% |
| Frost density | ±8.2% |

According to the theory of phase change kinetics, the phase change is related to the partial pressure of water vapor. Therefore, the partial pressure of water vapor is also used as a control variable. Furthermore, since the overall pressure of humid air is much larger than the partial pressure of water vapor, the working pressure around the cooling surface is approximated as the overall pressure in the cavity. For safety and operability reasons, the upper limit of high pressure in this experiment is set to $2.0 \times 10^5$ Pa. Within the range of working pressure around the cooling surface from $0.25 \times 10^5$ Pa to $2.0 \times 10^5$ Pa, a series of experimental frosting studies have been conducted. The temperatures of air and cooling surface and partial pressure of water vapor are adjusted to study their impacts on frost formation under different working pressures around the cooling surface. The experimental conditions are shown in Table 2.

**Table 2.** Test conditions in the three sets of frosting experiments.

| Set No. | Partial Pressure of Water Vapor $P_v$ [Pa] | Temperature of Air $T_a$ [°C] | Temperature of Cooling Surface $T_w$ [°C] |
|---------|---------------------|----------------|--------------------------------|
| A | 350<br>935<br>1400 | 20 | −38 |
| B | 350 | 15<br>20<br>25 | −38 |
| C | 350 | 20 | −20<br>−28<br>−38 |

## 3. Results and Discussion

### 3.1. Reliability Analysis

The stability of the cavity internal pressure is important to the test accuracy. For this reason, the internal pressure of the cavity during the experiment is recorded, as shown in Figure 3. The pressure in the cavity can be regarded as the working pressure around the cooling surface. The working pressure around the cooling surface is maintained by controlling the opening of the inlet and outlet valves of the cavity. The internal pressure of the cavity is maintained around the preset pressure.

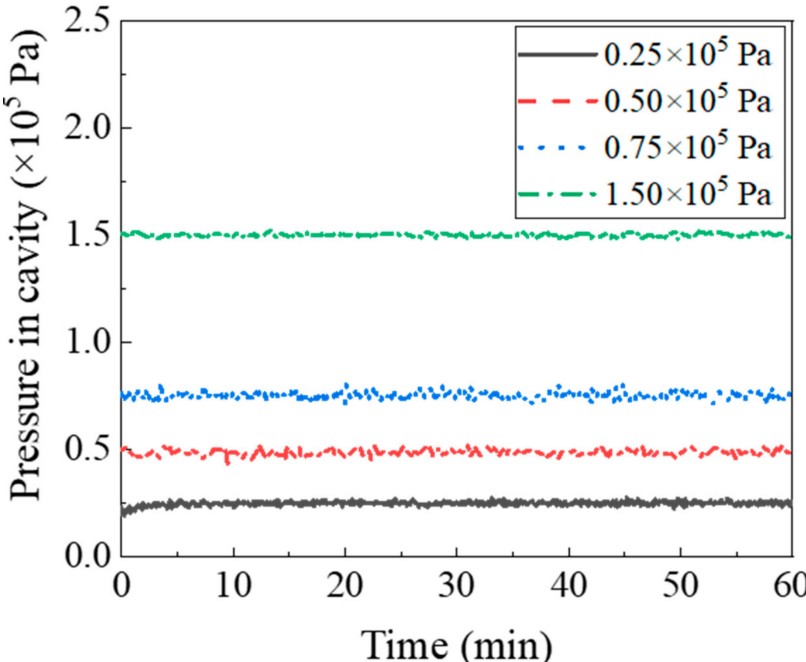

**Figure 3.** Transient working pressure in the cavity during the experiment.

For the purpose of obtaining the conclusions with high creditability, repeated tests are carried out. The selected working condition is $T_a$ = 20 °C, $P_v$ = 935 Pa, $T_w$ = −38 °C, and $P_a$ = 1.50 × 10$^5$ Pa. The frost thickness of the two experiments is shown in Figure 4. In the initial period, there is almost no difference between the frost thickness, while it increases to 0.07 mm at 60 min. The overall deviation is 3.2%, which is within the acceptable range.

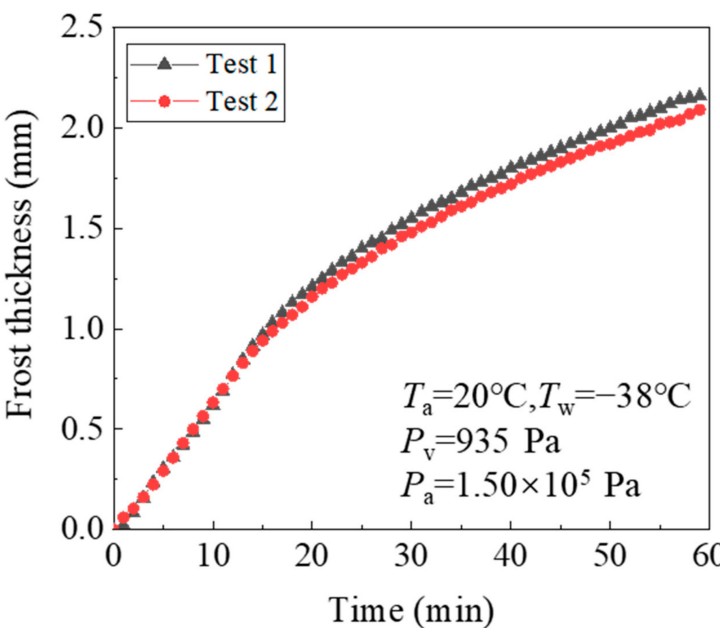

**Figure 4.** Frost thickness during the repeatability tests.

*3.2. Influence of Working Pressure around the Cooling Surface on Frost Crystal Morphology*

The frost crystal morphologies are photographed at 10 min, as shown in Figure 5, when $T_a$ = 20 °C, $T_w$ = −38 °C, $P_v$ = 350 Pa, and $P_a$ = 0.25 × 10$^5$ Pa, 0.75 × 10$^5$ Pa and 2.00 × 10$^5$ Pa, respectively. The frost crystal morphologies shift from feather to needle shape when the working pressure around the cooling surface increases, as shown in Figure 5b. The frost crystals at 0.25 × 10$^5$ Pa are thicker than those at higher pressure conditions. It can be attributed to the water vapor needing to cross a nucleation barrier when forming droplets or ice nuclei on cooling surfaces. Then the new frost nuclei can be created and continue to grow. The nucleation barrier is related positively to the overall air pressure, which affects the diffusion coefficient of water vapor directly [17]. Therefore, higher working pressure raises the nucleation barrier for forming frost nuclei, resulting in a slower growth rate of frost crystal. Moreover, the distribution of frost crystals in the lateral direction gradually becomes thinner with the increasing working pressure around the cooling surface.

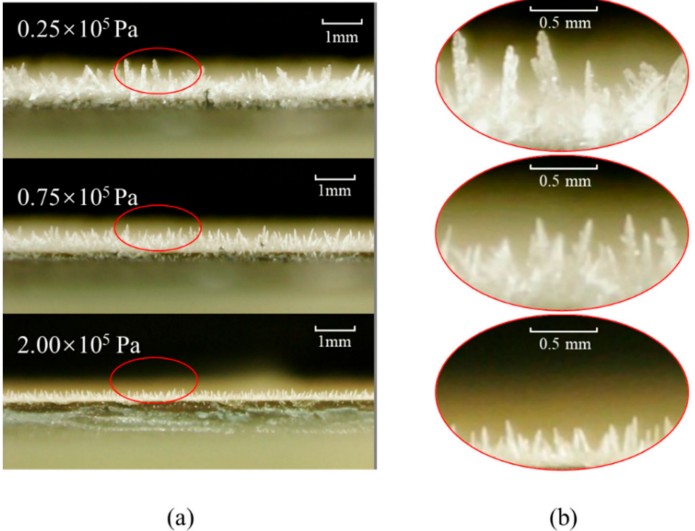

(a)                                                                                           (b)

**Figure 5.** (**a**) Morphologies and (**b**) local details of frost crystals under different working pressures around the cooling surface.

### 3.3. Influence of Working Pressure around the Cooling Surface on Frost Thickness

The growth habit of the frost layer is mainly reflected by the frost thickness, which is the most important and most intuitive parameter among the frost characteristics. The frost thickness under different working pressures around the cooling surface is shown in Figure 6. The frost thickness increases with time. Furthermore, the frost thickness decreases with rising working pressure around the cooling surface. The initial growth rate of frost thickness is slower when the working pressure around the cooling surface is higher, indicating that crystallization of frost is more difficult. It can be attributed to the higher pressure that leads to a stronger nucleation barrier of frost crystals, so the nucleation rate of frost crystals lessens. Furthermore, from the perspective of molecular diffusion, the growth rate of frost thickness is related to the concentration of water molecules around the frost crystals. The concentration of water molecules is proportional to the molecular diffusion coefficient [17]. In frosting, water vapor is continuously consumed, so a water vapor concentration gradient is formed near the frost crystal. With the decrease in pressure, the molecular diffusion coefficient augments. The water vapor diffuses to the frost crystal surfaces more easily. Therefore, the concentration difference of water vapor around the frost crystal surface is higher, which leads to faster growth of the frost crystals.

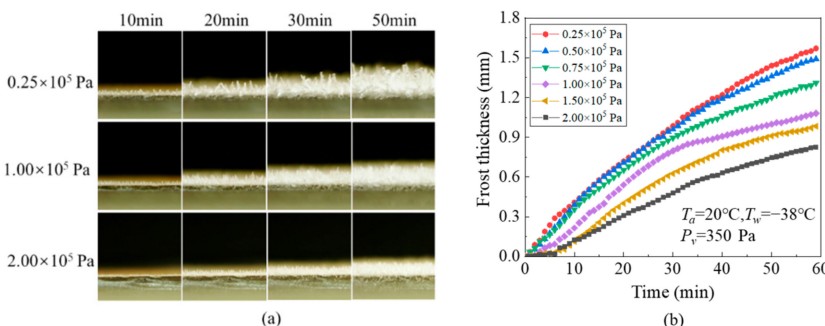

**Figure 6.** Frost thickness under different working pressures around the cooling surface with time: (**a**) Photos; (**b**) diagram of frost thickness.

### 3.4. Influence of Working Pressure around the Cooling Surface on the Frost Accumulation and Frost Density

The frost accumulation and frost density with different working pressures around the cooling surface at 60 min are shown in Figure 7. The frost accumulation reduces with a higher working pressure around the cooling surface. On the one hand, it can be attributed to the diminishing diffusion coefficient of water vapor, which means that less water vapor condenses into frost crystals. On the other hand, the thermodynamic potential energy driving any crystallization process needs to exceed the change in chemical potential from one phase to another. This potential energy can be either supercooling in the case of freezing or supersaturation in the case of condensation from water vapor [7]. When the temperatures of air and cooling surface keep constant, the surface temperature of the frost crystal is invariant. Therefore, its corresponding saturation vapor pressure is the same. When the working pressure near the frost crystal reduces, excess concentration of water vapor leads to a larger driving potential. The phase transition rate is in direct proportion to the driving potential. Therefore, more water vapor is deposited and condensed into frost crystals, which leads to an increase in frost accumulation. In addition, the growth rate of frost accumulation declines more rapidly than frost thickness with higher pressure, and the frost density reduces with pressure, as shown in Figure 7. Under conditions where other parameters are constant, the lateral growth of frost crystals is notably promoted when the working pressure around the cooling surface lessens (from Figure 5), resulting in more narrow gaps between the frost crystals, while the growth in the vertical direction is not significant. Therefore, the reduction of density is reflected through the pressure effect on the growth habit of frost crystal.

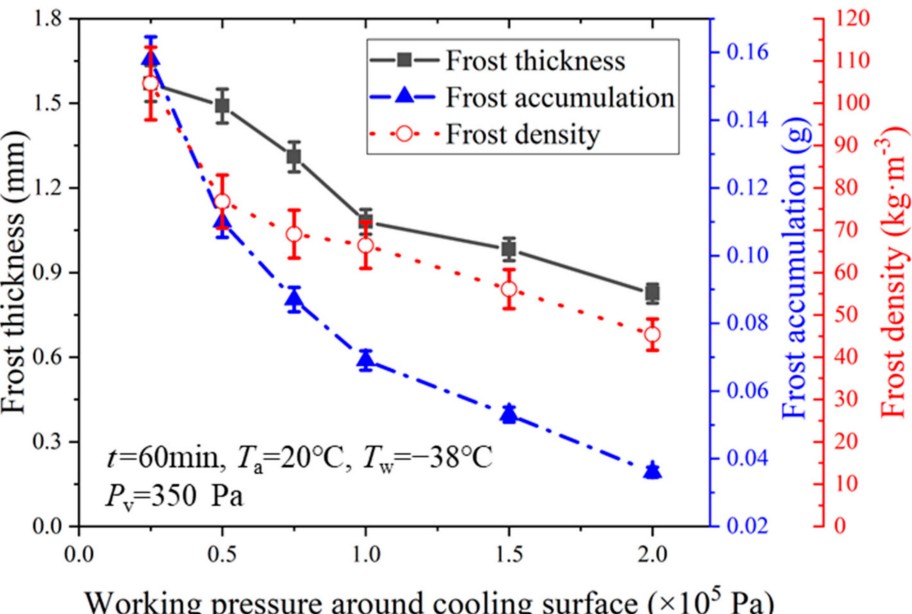

**Figure 7.** Frost characteristics with different working pressures around cooling surface at 60 min.

*3.5. Influence of Other Parameters*

3.5.1. Partial Pressure of Water Vapor in Humid Air

The concentration difference of water vapor is determined directly by the partial pressure of water vapor around the cooling surface, which is one of the important factors that determines the frosting crystallization. The frost crystal morphologies under different partial pressures of water vapor ($P_v$ = 350 Pa, 935 Pa, 1400 Pa) at 10 min are shown in Figure 8. The frost crystal morphologies become more abundant with a higher partial pressure of water vapor, changing from needle to feather shape. The lateral size of frost crystals gradually becomes stronger with the lower working pressure around the cooling surface due to the augmenting of diffusion coefficient of water vapor. Furthermore, the effect of partial pressure of water vapor on the lateral growth of frost crystals is greater than the effect of overall working pressure.

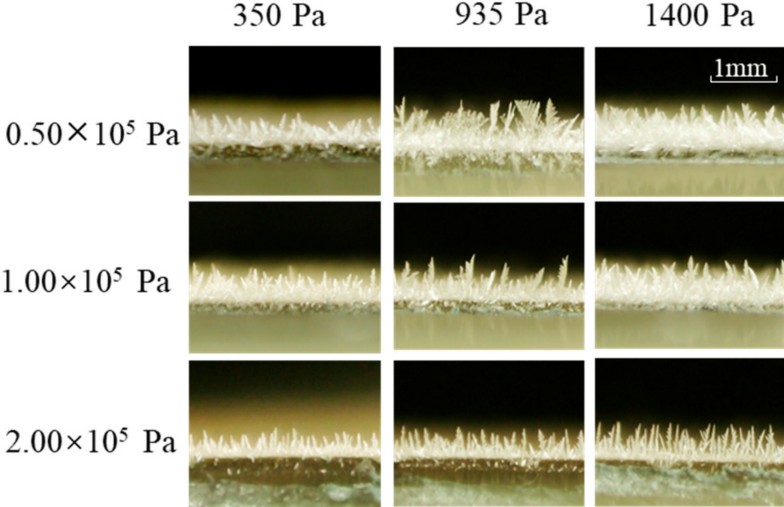

**Figure 8.** Frost crystal morphologies with working pressure around cooling surface under different partial pressures of water vapor at 10 min.

The frost thickness with working pressure around the cooling surface under different partial pressures of water vapor at 60 min is shown in Figure 9a. The frost thickness

decreases with the higher working pressure around the cooling surface. In addition, the frost thickness increases with the higher partial pressures of water vapor, which can be attributed to the larger concentration difference of water vapor on the frost crystal surface. The frost accumulation and density at 60 min rise with the partial pressure of water vapor due to the enhanced driving force of mass transfer, as shown in Figure 9b,c. This rise is weakened with the higher working pressure around the cooling surface. Comparing the absolute differences of frost thickness, frost accumulation and density from low to high pressure ($0.5 \times 10^5$ Pa to $2.00 \times 10^5$ Pa), they lessen from 1.43 mm to 0.89 mm, 0.247 g to 0.120 g and 50.48 kg/m$^3$ to 49.34 kg/m$^3$, respectively, when the partial pressure of water vapor increases from 350 Pa to 1400 Pa.

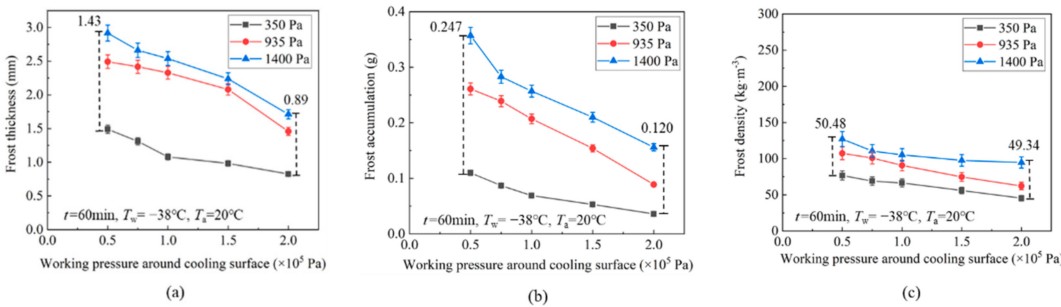

**Figure 9.** (**a**) Frost thickness; (**b**) frost accumulation; (**c**) frost density with working pressure around cooling surface under different partial pressures of water vapor at 60 min.

### 3.5.2. Temperature of Humid Air

The temperature of humid air affects the temperature field around the frost crystal surface, which is a major parameter affecting the frost crystal morphology. The frost crystal morphologies under different air temperatures ($T_a$ = 15 °C, 20 °C, 25 °C) at 10 min are shown in Figure 10. With a cooler air temperature, the motion of water vapor is so less violent that there is enough time to develop the frost crystal. Therefore, the growth of the frost crystals is promoted with a dendritic growth on the crystal prism, and the frost crystals become stronger. The frost crystal morphologies show a needle–columnar–feather shape transition with a cooler air temperature. In addition, it is difficult to distinguish individual frost crystals at a higher air temperature. The growth of frost crystals is also enriched with a lower working pressure around the cooling surface.

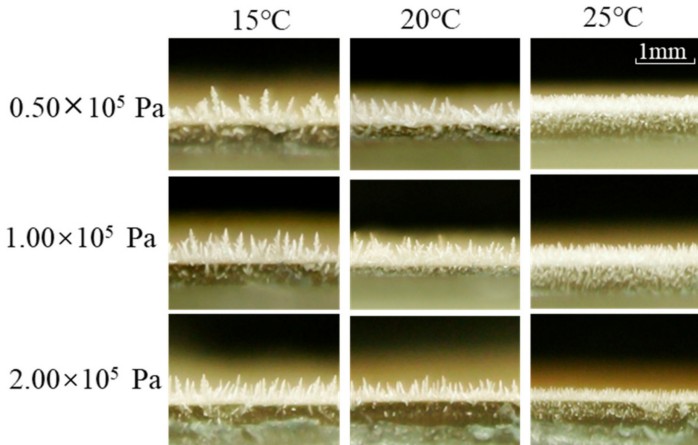

**Figure 10.** Frost crystal morphologies with working pressure around cooling surface under different temperatures of humid air at 10 min.

The frost thickness, at 60 min, with working pressure around the cooling surface under different temperatures of humid air is shown in Figure 11a. The frost thickness decreases

with higher pressure and warmer air temperature. The surface temperature of the frost layer rises in direct proportion to the air temperature, which hinders the frost growth. The frost accumulation, at 60 min, with working pressure under different temperatures of humid air is shown in Figure 11b. The frost accumulation increases with higher air temperature. When the air temperature rises, the heat transfer is enhanced due to the expansile temperature difference. According to the Lewis heat and mass transfer analogy theory, mass transfer is positively related to the amount of heat transfer. From Figure 11b,c, the growth rate of frost accumulation and frost layer density are augmented with higher air temperature. This enhancement is reduced as the working pressure increases. Comparing the absolute differences of frost thickness, frost accumulation and density from low to high pressure ($0.25 \times 10^5$ Pa to $2.00 \times 10^5$ Pa), they lessen from 0.292 mm to 0.272 mm, 0.134 g to 0.009 g and 108.73 kg/m$^3$ to 31.49 kg/m$^3$, respectively, when the temperature of humid air increases from 15 °C to 25 °C.

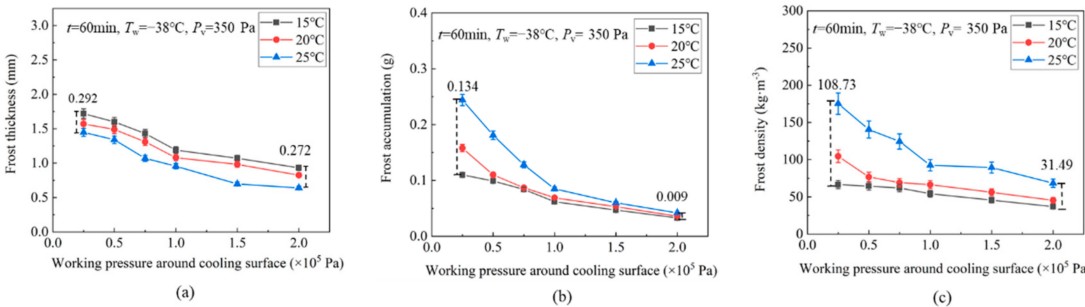

**Figure 11.** (**a**) Frost thickness; (**b**) frost accumulation; (**c**) frost density with working pressure around cooling surface under different temperatures of humid air at 60 min.

### 3.5.3. Temperature of Cooling Surface

The temperature distribution inside the frost crystals is mainly affected by the temperature of the cooling surface, which determines the potential barrier in the initial period of frost nucleation. The frost crystal morphologies under different temperatures of cooling surfaces ($T_w$ = −20 °C, −28 °C, −38 °C) at 10 min are shown in Figure 12. When the temperature of the cooling surface increases, the frost crystals are hindered in a vertical direction due to the insufficient driving force of frosting, whereas they become stronger in the lateral direction. The top surface of the frost crystal becomes rounded and blunt. The change is very obvious from needle to columnar shape in lower working pressure ($0.50 \times 10^5$ Pa) around the cooling surface. In addition, the larger diffusion coefficient of water vapor leads to obvious lateral growth with the decrease in working pressure around the cooling surface.

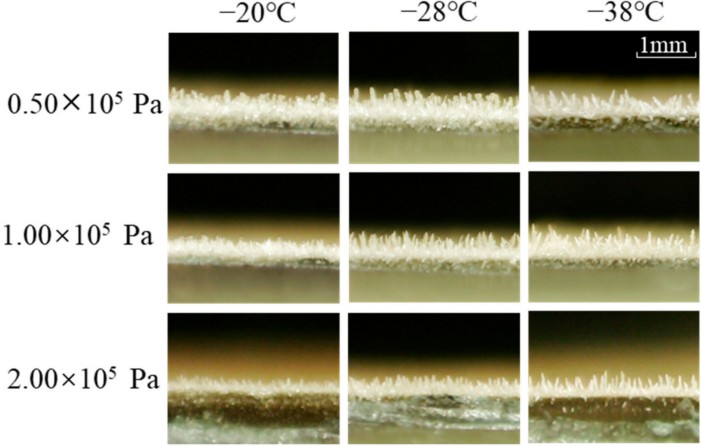

**Figure 12.** Frost crystal morphologies with working pressure around cooling surface under different temperatures of cooling surfaces at 10 min.

The frost thickness and frost accumulation under different temperatures of cooling surfaces at 60 min are shown in Figure 13a,b. The frost thickness and frost accumulation decrease with the higher temperature of the cooling surface. When the temperature of the cooling surface increases from $-38\,°C$ to $-20\,°C$, the surface temperature of frost crystal rises. The heat transfer is deteriorated with the higher temperature of the cooling surface due to a smaller temperature difference. It means that fewer water vapor molecules diffuse into the surface of the frost crystal, making it more difficult to form the frost branches which create more gaps between the frost crystals. Therefore, the frost density increases with a higher temperature of the cooling surface, as shown in Figure 13c. Comparing the absolute differences of frost thickness, frost accumulation, and density from low to high pressure ($0.25 \times 10^5$ Pa to $2.00 \times 10^5$ Pa), they lessen from 1.00 mm to 0.55 mm, 0.043 g to 0.011 g and 105.75 kg m$^{-3}$ to 48.29 kg m$^{-3}$, respectively, when the temperature of the cooling surface increases from $-38\,°C$ to $-20\,°C$.

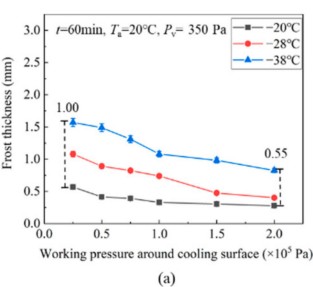 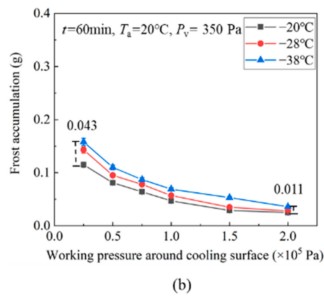 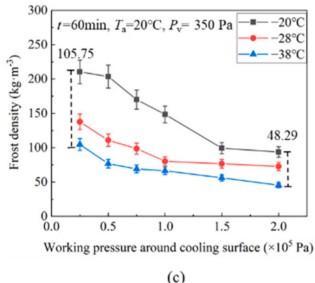

(a)  (b)  (c)

**Figure 13.** (**a**) Frost thickness; (**b**) frost accumulation; (**c**) frost density with working pressure around cooling surface under different temperatures of cooling surfaces at 60 min.

### 3.6. Dimensionless Correlation of Frost Thickness

An appropriate empirical correlation is helpful to understand and predict the relationship between frost characteristics and various parameters. Owing to the complexity of frosting where multiple parameters are changing, an empirical correlation that applies to most environments is hard to obtain. Compared with the correlations proposed by most other frost studies [25–27], the dimensionless correlation between frost thickness and working pressure around the cooling surface is proposed in this study. The dimensionless process takes place among the frost thickness, temperature of air, temperature of cooling surface, partial pressure of water vapor, and working pressure around the cooling surface, as shown in Table 3. The dimensionless correlation formula of the frost thickness is provided in Equation (2). The available range of frost thickness in the correlation equation is: $T_a = 15\sim25\,°C$, $T_w = -38\sim-10\,°C$, partial pressure of water vapor $P_v = 350\sim1400$ Pa, working pressure around the cooling surface $P_a = 0.15 \times 10^5\sim2.00 \times 10^5$ Pa.

$$y* = 8.180 \times 10^{-17} \, T_a*^{-6.115} \, T_w*^{7.427} \, P_v*^{0.519} \, P_a*^{-0.326} \, Gr^{4.353} \, (Fo^{0.38} + 1)^{1.558} \qquad (2)$$

**Table 3.** Dimensionless variables for correlation formula.

| Dimensionless Variable | $y*$ | $T_a*$ | $T_w*$ | $P_v*$ | $P_a*$ |
|---|---|---|---|---|---|
| Normalized process | $y/l$ | $T_a/273.15$ | $T_w/273.15$ | $P_v/101325$ | $P_a/101325$ |

The fit of the correlated data increases with test data as frost growth in this correlation. The overall error of the dimensionless frost thickness between the correlated data and the test data is shown in Figure 14. The overall error is within $\pm25\%$. The errors in the experiment mostly come from photographing and parametric measurements. In the early frosting period, the small deviation produces a large relative error due to the small frost thickness (brown data), whereas the overall error is still within a reasonable range.

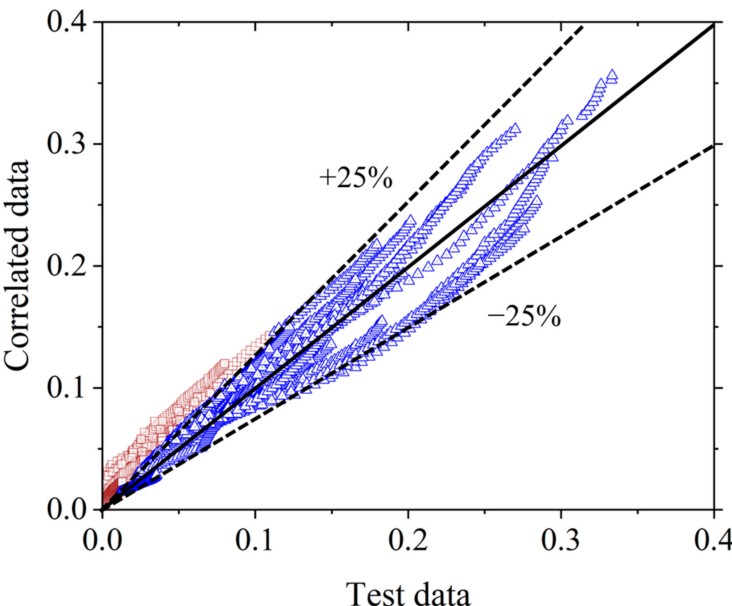

**Figure 14.** Comparison of test and correlated data for dimensionless frost thickness.

## 4. Conclusions

In this paper, the effect of working pressure around the cooling surface on frosting is investigated through a frosting visualization test rig. The frost crystal morphologies and frost layer characteristics under different working pressures around the cooling surface are studied with partial pressure of water vapor, temperature of humid air, and temperature of cooling surface. The main conclusions are as follows.

1.  The vertical growth of frost crystals and the lateral distribution of frost crystals are strongly influenced by working pressure around the cooling surface. In a lower pressure environment, the frost crystals grow faster in the vertical direction and are more densely distributed in the lateral direction. The lateral growth habits of frost crystal are a primary function of air temperature and partial pressure of water vapor.

2.  The frost crystal morphologies are more various with the variation of influencing parameters in a lower pressure environment. Frost crystals transform from needle to feather shape with the rising partial pressure of water vapor. A feather–columnar–needle shape transition of frost crystals is presented with hotter air temperature. In addition, the needle frost crystals lose their sharp point and become a clear columnar with a higher temperature of the cooling surface. However, under the higher pressure condition, only insignificant frost crystal morphologies shift between the needle and columnar shape with the variation of other parameters.

3.  Under low working pressure around the cooling surface, the absolute differences of frost thickness, frost accumulation and density with different environmental parameters are larger than those at high pressures. Under different working pressures around the cooling surface, the dominating factor in the growth of the frost layer is presented by the partial pressure of water vapor. The dimensionless correlation equations of the frost thickness within a certain range on the temperature of humid air, temperature of cooling surface, partial pressure of water vapor, and working pressure around the cooling surface are proposed. The overall relative error is within $\pm$ 25%.

**Author Contributions:** Conceptualization, T.L. and X.L.; methodology, T.L. and X.L.; validation, X.L. and X.D.; formal analysis, X.L. and X.D.; investigation, X.L., M.Q. and S.Y.; resources, M.Q. and S.Y.; data curation, S.Y. and X.L.; writing—original draft preparation, X.L.; writing—review and editing, T.L., X.L. and Y.H.; visualization, T.L. and X.L.; supervision, T.L. and Y.H.; project administration, T.L.; funding acquisition, T.L. All authors have read and agreed to the published version of the manuscript.

**Funding:** This project is supported by the National Natural Science Foundation of China (51976150), the Fundamental Research Funds for the Central Universities, and the Youth Innovation Team of Shaanxi Universities.

**Institutional Review Board Statement:** Not applicable.

**Informed Consent Statement:** The study did not involve humans.

**Conflicts of Interest:** The authors declare no conflict of interest.

## Abbreviations

| | |
|---|---|
| $A$ | plate area (m$^2$) |
| $Fo$ | Fourier number |
| $Gr$ | Grashof number |
| $l$ | characteristic length (m) |
| $m$ | frost accumulation (kg) |
| $P_a$ | working pressure around the cooling surface (Pa) |
| $P_{v4.6}$ | partial pressure of water vapor (Pa) |
| $T_a$ | temperature of air (°C) |
| $T_w$ | temperature of cooling surface ((°C) |
| $y_f$ | average frost thickness (m) |
| $\rho_f$ | average density (kg m$^{-3}$) |

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
