# Peer review of "Experimental Study on Frost Crystal Morphologies and Frosting Characteristics under Different Working Pressures"

_applsci, doi:10.3390/app12084025_

Round 1

Reviewer 1 Report

In this manuscript, Tianwei Lai et al. study the influence of the overall pressure of vapor on frost formation by carefully controlling the other influencing factors. The whole experiment design is credible and the results are scientifically sound. I would consider it to be accepted by Applied Sciences with minor modifications by answering my questions properly.

  1. The mass of frost is very important in the paper while the authors didn't give clear details and how to determine the uncertainty of 4.16% in Table 1. (Line 140)
  2. Is there any thermocouple hanging in the chamber to get the vapor temperature? If not, it's a good improvement for the device to assure the temperature stability of vapor after heat exchange in each experiment.
  3. What are the dimensions of the growth area?
  4. I notice that for the morphology photos in e.g. Fig. 5,6,8, the substrates of frost formation are different from each other. Are they just different regions of the cold head? Why don't you choose the same region of the cold head to assure the same surface condition?
  5. And in how large an area the morphology of frost won't change by fixing all parameters? If it's not large enough, I may suspect the experimental uncertainty will increase a lot.
  6.  Can authors add error bars in the figures?

Reviewer 2 Report

It is a well written interesting paper. All of the experimental work was planned carefully. The presentation of the article is adequate. 

This is an interesting paper about an important question from application and basic science points too. It is very well and clearly written. All of the experimantal part is carefully planned and the prsentation (graph, picture ..) is also nice I got a clear simple answer for all of the questions, which answer are properly presented.

I have only on minor comment.

I believe I understand the proper meaning the following  part of a sentence "stronger vertically .. (in conclusion)" Please try to use different more precise description. 
